# Global climate forcing on late Miocene establishment of the Pampean aeolian system in South America

Blake Stubbins [1], Andrew L. Leier [1], David L. Barbeau Jr.[1], Alex Pullen [2] ✉, Jordan T. Abell [3,4], Junsheng Nie [5], Marcelo A. Zárate[6] & Mary Kate Fidler[2]

Wind-blown dust from southern South America links the terrestrial, marine, atmospheric, and biological components of Earth's climate system. The Pampas of central Argentina (~33°–39° S) contain a Miocene to Holocene aeolian record that spans an important interval of global cooling. Upper Miocene sediment provenance based on $n = 3299$ detrital-zircon U-Pb ages is consistent with the provenance of Pleistocene–Holocene deposits, indicating the Pampas are the site of a long-lived fluvial-aeolian system that has been operating since the late Miocene. Here, we show the establishment of aeolian sedimentation in the Pampas coincided with late Miocene cooling. These findings, combined with those from the Chinese Loess Plateau (~33°–39° N) underscore: (1) the role of fluvial transport in the development and maintenance of temporally persistent mid-latitude loess provinces; and (2) a global-climate forcing mechanism behind the establishment of large mid-latitude loess provinces during the late Miocene.

The Pampas of central Argentina, South America (Fig. 1) record the production, transport, and deposition of dust in a region where wind-blown detritus has been shown to influence several components of the Earth's climate system[1–5]. Lithogenic dust originating from South America alters radiative forcing budgets[1,6] and provides micronutrients essential to photosynthetic organisms in the ocean and terrestrial sites that are capable of sequestering atmospheric $CO_2$[7–10]. In particular instances, aeolian detrital material serves as an important component of an internal feedback system: once initiated, arid, wind-dominated transport systems can be self-perpetuating, creating conditions that reinforce aridification and yield greater effluxes of aeolian dust[11,12]. Considering the potential ramifications for regional and global climate, reconstructing the long-term history of southern South American dust dynamics is critically important for understanding Earth's climatic evolution from the relatively warm climate of the mid-Miocene to the glacial-interglacial cycles of the Quaternary.

The Pleistocene–Holocene history of aeolian transport and deposition on the Argentine Pampas is recorded by ~700,000 km² of aeolian sand and loess between ~30°–39° S latitude (Fig. 1)[13]. Miocene surface uplift of the Andes set the conditions for sediment supply to the lowlands and altered basin hydrology[14,15]. Most of the sediments exposed at the surface today accumulated during the late Pleistocene[16], when large volumes of sediment from glaciated Andean watersheds were conveyed to the foreland by regional river systems[17,18], which coupled with arid conditions in the Pampas, created ideal conditions for aeolian transport by westerly and southwesterly winds[19–22].

Upper Miocene strata in the southwestern Pampas record pre-Quaternary aeolian transport and deposition. Globally, the late Miocene is associated with declining atmospheric $pCO_2$[23], expansion of C4 plants[24,25], declining sea surface temperatures, and increasing meridional sea surface temperature gradients[26]. Increased aridity and

[1]School of the Earth, Ocean and Environment, University of South Carolina, Columbia, SC 29208, USA. [2]Department of Environmental Engineering and Earth Sciences, Clemson University, Clemson, SC 29634, USA. [3]Department of Geosciences, University of Arizona, Tucson, AZ 85721, USA. [4]Department of Earth and Environmental Sciences, Lehigh University, Pennsylvania, PA 18015, USA. [5]Key Laboratory of Western China's Environmental Systems (Ministry of Education), College of Earth and Environmental Sciences, Lanzhou University, Lanzhou 730000, China. [6]Instituto de Ciencias de la Tierra y Ambientales de La Pampa, CONICET Universidad Nacional de La Pampa, La Pampa, Argentina. ✉e-mail: apullen@clemson.edu

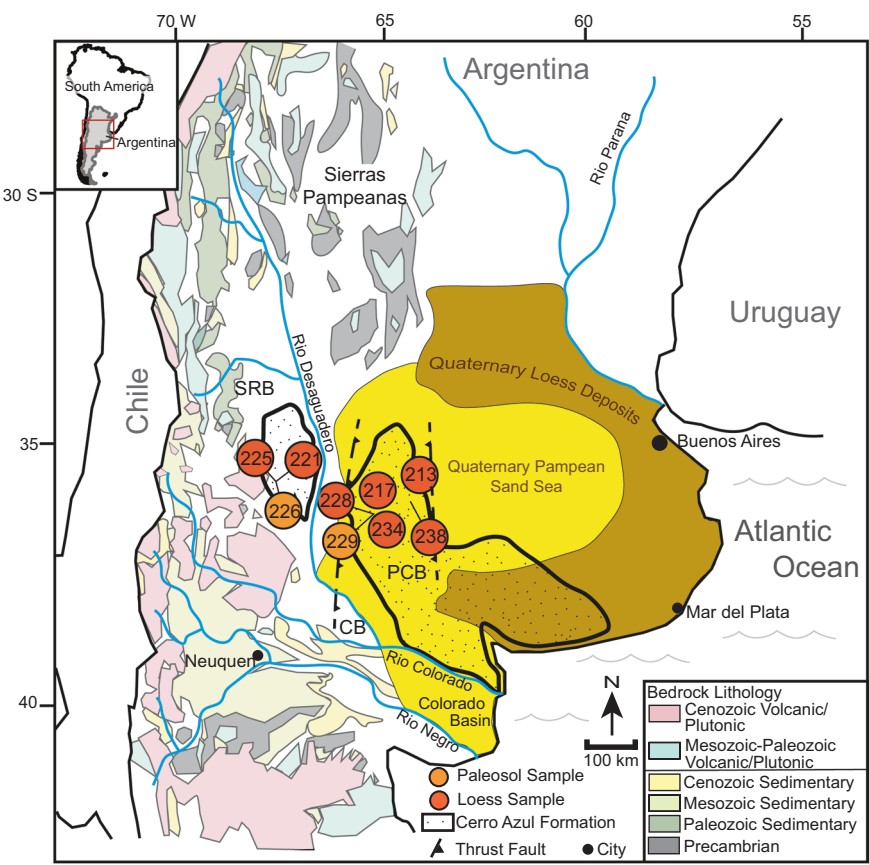

**Fig. 1 | Simplified geologic map of the Pampas and surrounding areas in South America.** The map includes Quaternary loess and sand deposits, the distribution of the upper Miocene Cerro Azul Formation (stippled region), and sample locations. Circles depict specific locations of loess and paleosol samples from the Cerro Azul Formation. Major regional rivers, including the Negro, Colorado, Desaguadero, and Paraná rivers are shown. Abbreviations are as follows, Chadileuvú block (CB), Pampa Central block (PCB), and San Rafael block (SRB). Quaternary sediment was transported from the Cordillera as well as uplifted foreland blocks through the Colorado, Negro, and Desaguadero rivers and entrained by westerly and southwesterly winds. Paleoclimate data referenced in the text were collected from the Sierras Pampeanas.

seasonality in the Andes during the late Miocene is linked to increased sediment production and deposition of aeolian strata[27–29].

In the southwestern Pampas, the late Miocene climate is recorded by loess and loessic paleosols in the upper Miocene Cerro Azul Formation[30]. U-Pb detrital zircon data from the Cerro Azul Formation demonstrate that the fluvial-aeolian system responsible for upper Pleistocene–Holocene aeolian deposits on the Pampas was established and operating during the late Miocene, millions of years before the onset of latest Pliocene-Pleistocene glaciations. The fluvial-aeolian framework that has directed dust production, transport, and deposition in the Pampas consists of multiple components including sediment sources, foreland rivers and climate/atmospheric conditions conducive to aeolian transport. We propose this system has been in place since the late Miocene, responding to climatic drivers and orographic changes in the Andes, as well as drainage reorganization in the Andean foreland—a mechanism similar to that of the mid-latitude Chinese Loess Plateau in the Northern Hemisphere[31–33].

The Pampas of central Argentina are a vast, low-relief expanse within the Andean foreland basin that extend from the foothills of the Andes in the west to the South Atlantic Ocean in the east, and the Sierras Pampeanas in the north and the Colorado Basin in the south Fig. 1[34]. Surficial deposits on the Pampas comprise the most extensive aeolian depositional system in South America, including an aeolian sand sea and a surrounding loess belt (~500 km long, 200 km wide) along its northern and eastern margins[13,17,35,36]. The majority of Pleistocene–Holocene aeolian deposits are vegetation-stabilized under current (interglacial) conditions[37]. Provenance data indicate these aeolian sediments were derived primarily from floodplain deflation of the Desaguadero, Colorado, and Negro rivers with additional input of volcaniclastic material from the Andean volcanic arc[17,35,38–40]. The present-day drainage configuration in this portion of the Andean foreland was generated as rock uplift migrated southward in the late Miocene to Pliocene, starting in the Sierras Pampeanas (26°–34° S) and moving south to include the Pampa Central block (35°–38° S) by the latest Miocene[15,29,41]. This resulted in topographic partitioning of the foreland. This partitioning caused the diversion of east-draining rivers southward, if present (e.g., ref. 39), and routed sediment to the Colorado Basin and the Atlantic coast between the Pampas and northern Patagonia[39,42]. The provenance of aeolian sediments deflated from megafans, bajadas (i.e., coalesced alluvial fans), and floodplains within the broken foreland has evolved in some areas in response to ongoing tectonics and changing catchment patterns[39,43].

Along the southern and western margins of the Pampas, the Pleistocene–Holocene aeolian deposits are underlain by the upper Miocene Cerro Azul Formation (Fig. 1)[30,41]. The Cerro Azul Formation was deposited between ca. 8.9 and ca. 5.5 Ma based on the presence of Chasicoan and Huayquerian faunas across outcrops that delineate the late Miocene portion of the South America Land Mammal Ages/Stage framework[30,41,44]. The deposits consist of sandstone and siltstone; fluvial sandstones and paleosol horizons in the lower half of the formation are replaced by loess and paleosol deposits with pedogenic carbonate nodules and rhizoliths in the upper half[30,45,46]. The distribution of facies and stratigraphic thicknesses of the Cerro Azul Formation suggest deposition in the back-bulge depozone of the

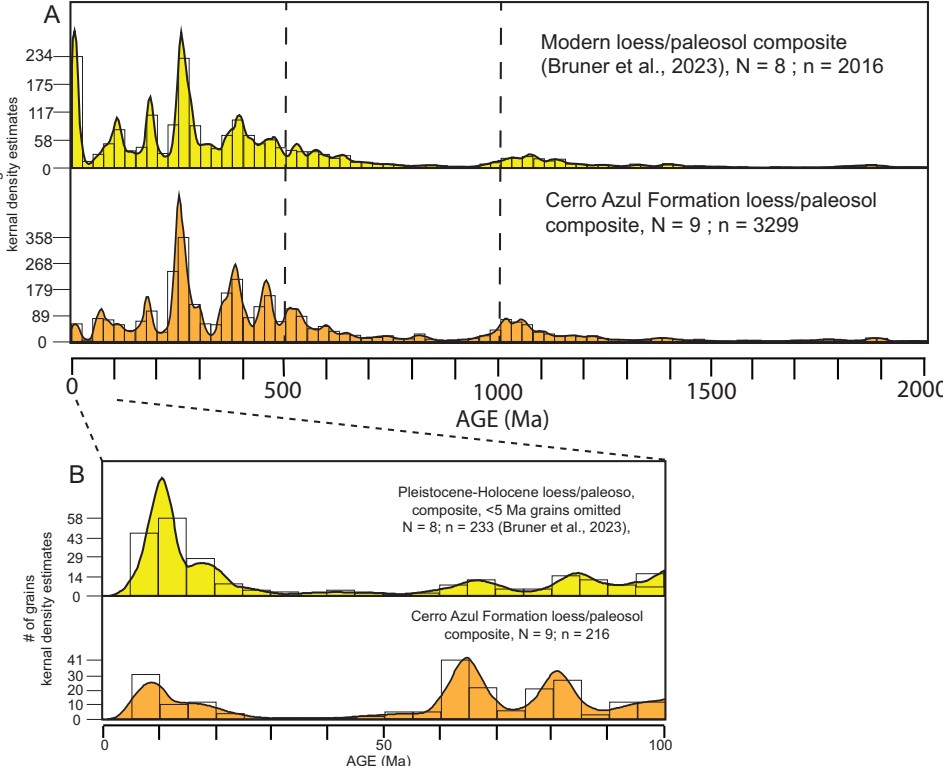

**Fig. 2 | Kernel density estimates (KDE) and histograms of U-Pb detrital zircon ages from the Cerro Azul Formation and Pleistocene–Holocene deposits.**
**A** Composite KDE of the Cerro Azul Formation (this study) and Pleistocene–Holocene loess and loessic paleosol deposits in the Pampas[40]. The Cerro Azul Formation and the Pleistocene–Holocene deposits contain the same detrital zircon age modes, with several of the largest age modes having similar proportions in both datasets. KDE are area-normalized and constructed with an Epanechnikov kernel using a bandwidth of 15 Myr. The number of grains within each 25 Myr histogram bin is shown on the y-axis. Fifty grains have ages older than 2 Ga and do not appear on the plot. **B** Composite KDE of the Cerro Azul Formation and the Pleistocene–Holocene deposits, excluding ages <5 Ma, which post-date deposition of the Cerro Azul Formation. KDEs are area-normalized and constructed with an Epanechnikov kernel using a bandwidth of 5 Myr. The base image is from Google Earth Pro using Landsat: Copernicus.

Andean retroarc foreland basin prior to uplift of the Pampa Central block, which segmented the foreland between ~35° S and ~38° S during the latest Miocene[41]. Aeolian deposits in the Cerro Azul Formation, recorded as loess-paleosol sequences, represent the oldest aeolian deposits in the Pampean Neogene succession and suggest generally more arid conditions in the late Miocene relative to the middle Miocene[45]. Greater aridity and/or seasonality during the late Miocene at 35°–38° S is consistent with rapid ecological and hydrologic changes across central South America during late Miocene cooling that followed the Middle Miocene Climatic Optimum[47,48].

**Table 1 | Calculations of maximum depositional ages based on U-Pb detrital zircon ages using various approaches**

| Sample | n | MLA | YSG | Y3Z | YPP |
|---|---|---|---|---|---|
| 21AR213 | 520 | 6.09 ± 0.28 | 4.6 ± 0.7 | 14.05 ± 2.29 | 6.04 |
| 21AR217 | 247 | 7.05 ± 0.7 | 6.2 ± 0.9 | 6.64 ± 0.95 | 6.6 |
| 21AR221 | 187 | 6.78 ± 0.56 | 6.5 ± 0.9 | 6.74 ± 0.77 | 6.74 |
| 21AR225 | 261 | 7.93 ± 0.59 | 7.5 ± 0.5 | 62.70 ± 4.52 | 7.43 |
| 21AR226 | 198 | 4.43 ± 0.17 | 4.4 ± 0.1 | 7.2 ± 1.2 | 4.31 |
| 21AR228 | 541 | 49.23 ± 2.28 | 18.6 ± 0.8 | 51.6 ± 5.4 | 52.15 |
| 21AR229 | 287 | 9.43 ± 0.84 | 8.4 ± 0.7 | 61.7 ± 6.6 | 66.53 |
| 21AR234 | 521 | 5.94 ± 0.41 | 5.0 ± 0.5 | 6.0 ± 0.7 | 5.97 |
| 21AR238 | 537 | 9.05 ± 0.64 | 5.2 ± 0.8 | 18.8 ± 2.8 | 11.18 |

Ages reported in millions of years.
*MLA* Maximum likelihood age (Vermeesch, 2021), *YSG* Youngest single grain (2σ uncertainty), *Y3Z* Youngest three grains with overlapping 2σ uncertainty, calculated with 2σ uncertainty, *YPP* Youngest graphical peak.

## Results

We collected $N = 9$ samples of loess and loessic paleosol from exposures of the upper Cerro Azul Formation across a ~300 km east-west distance (Fig. 1) and analyzed these using U-Pb detrital zircon geochronology. Results are presented in Fig. 2 as histograms and kernel density estimate (KDE) plots, which display a smoothed distribution of detrital zircon ages from the Cerro Azul Formation. Sample locations, individual sample KDE plots, tabulated records, isotopic ratios, elemental concentration data, and uncertainties are included in the Supplementary Materials (Fig. S1 and Tables S1, S2). Maximum depositional ages were calculated using the maximum likelihood age algorithm[49] and yield ages generally between ca. 9 and 6 Ma (Table 1). Although these ages represent the maximum possible depositional age and not necessarily the actual depositional age, they are consistent with existing age constraints for the upper half of the Cerro Azul Formation[30,44].

All samples contain eight U-Pb zircon age populations, or modes, that are shared between samples and across locations (Fig. 2 and S1). Most of the zircon grains likely represent some degree of recycling; however, their U-Pb ages are consistent with sediment source rocks in the region. The eight age modes are: [1] 5–25 Ma, derived from the Andean magmatic arc[50]; [2] 50–125 Ma, derived from the Andean magmatic arc, including the North Patagonian batholith, or more specifically, megafans sourced from this area[51]; [3] 150–200 Ma, derived from the Chon Aike Silicic Large Igneous Province[52]; [4] 220–280 Ma, derived from the Choiyoi Magmatic Province[53]; [5] 330–400 Ma, derived from rocks associated with a Devonian-Carboniferous magmatic arc exposed in the San Rafael and Chadileuvú blocks[54]; [6] 420–475 Ma, derived from plutonic rocks associated

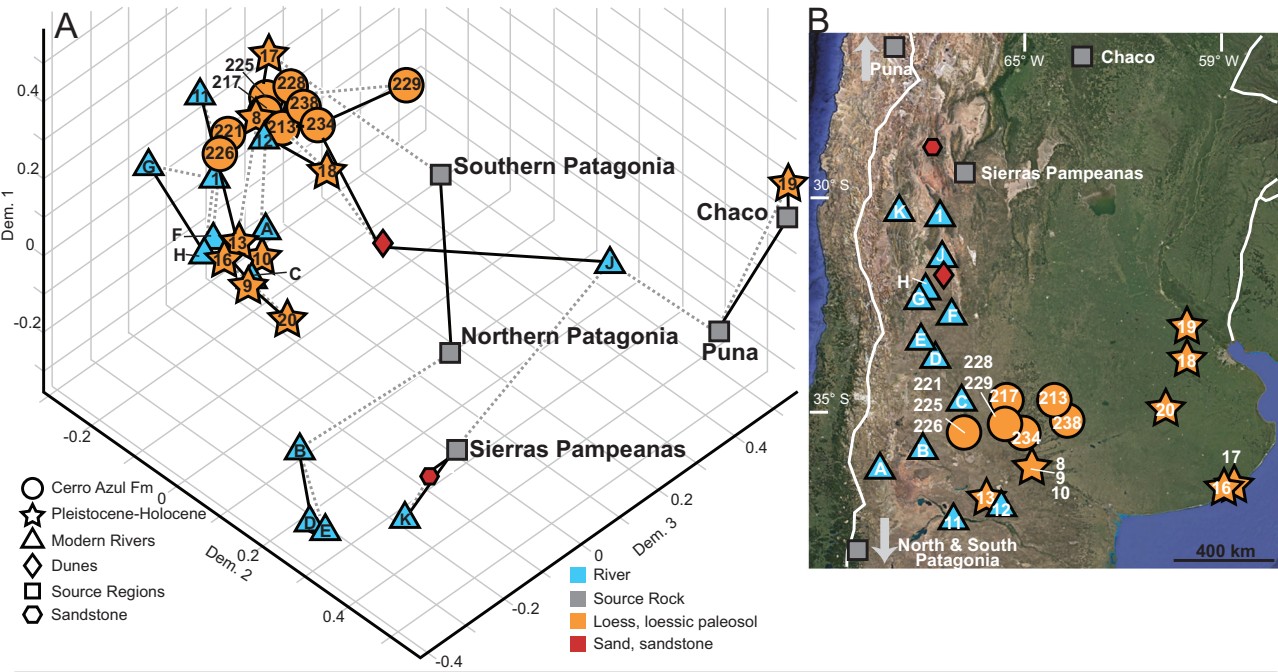

River samples: 1 = Rio Desaguadero; 11 = Rio Negro; 12 = Rio Colorado; A = Rio Neuquén; B = Rio Agua Escondida; C = Rio Desaguadero; D = Rio Atuel;
E = Rio Diamente; F = Rio Tunuyan; G = Rio Mendoza; H = Rio San Juan; J = Rio Bermejo; K = Rio Jachal.
Dunes, Holocene sand dunes. Sandstone = Miocene sandstone; Source regions = Sierras Pampeanas; Chaco; North and South Patagonian Andes; Puna.

**Fig. 3 | Three-dimensional multidimensional scaling (MDS) plots of detrital zircon data. A** Samples reported in this study were plotted with Pleistocene–Holocene loess and loessic paleosols, river samples, Holocene dunes, Miocene sandstones, and potential sediment source regions. Solid lines connect samples to their closest neighbor and dashed gray lines represent the second closest neighbor. The overlap between upper Miocene and Pleistocene–Holocene samples indicates statistical similarity and similar sediment provenance. Data sources: 1, 11, 12 from ref. 40; A-F from ref. 39; G-K from ref. 50; Dunes from ref. 78; Sandstone from ref. 15; Sierras Pampeanas from refs. 50,79–82; Chaco from refs. 83,84; North and South Patagonia from refs. 85,86; Puna from refs. 87–92. **B** Geographic location (from Google Earth; earth.google.com/web) of samples used in (**A**).

with the Famatinian arc[55]; [7] 480–540 Ma, derived from igneous rocks of the Pampean Orogeny and early magmatic activity of the Famatinian Orogeny[56]; and [8] 900–1200 Ma, derived from rocks of the Cuyania terrane, exposed west of the present-day Desaguadero basin[56].

The relative abundance of each population varies between samples, although the 220–280 Ma population is the largest age mode for all nine samples, with the 330–400 Ma mode being the second largest for all samples. When combined into a single KDE plot (Fig. 2), the relative abundance of each of the detrital zircon age modes are, from largest to smallest: [1] 220–280 Ma; [2] 330–400 Ma; [3] 900–1200 Ma; [4] 420–475 Ma; [5] 480–540 Ma; [6] 50–125 Ma; [7] 150–200 Ma; and [8] 5–25 Ma.

## Discussion

The combined detrital-zircon U-Pb ages from the upper Miocene Cerro Azul Formation are consistent with those from overlying Pleistocene–Holocene aeolian strata (Figs. 2, 3 and S1). Statistically, a comparison between the U-Pb ages of the two datasets yields a similarity coefficient of 0.91 and a cross-correlation coefficient of 0.55[57]. The eight detrital zircon age modes in the Cerro Azul Formation are present in the KDE of the Pleistocene–Holocene aeolian strata of the Pampas, suggesting derivation from the same sources (Fig. 2)[40]. Moreover, several of the age modes have similar relative abundances, with the 220–280 Ma population representing the largest percentage of zircons in the samples (18% of total grains in Pleistocene–Holocene deposits, 19% in the Cerro Azul Formation). The largest difference between the U-Pb data of the Cerro Azul Formation and the Pleistocene–Holocene deposits is the greater abundance of detrital zircon grains with ages of 0–25 Ma in the Pleistocene–Holocene sediments (Fig. 2). However, a significant proportion of the 0–25 Ma age mode in the Pleistocene–Holocene strata have ages between 0 and 5 Ma (~40%) owing to renewed Pliocene-to-present volcanism[58], which postdates deposition of the Cerro Azul Formation. Excluding zircons from the Pleistocene–Holocene dataset that crystallized after deposition of the Cerro Azul Formation results in greater similarity between upper Miocene and the Pleistocene–Holocene deposits, with a similarity coefficient of 0.96 and a cross-correlation coefficient of 0.71, indicating similar sediment sources.

The nonparametric multidimensional scaling (MDS) statistic provides additional evidence of similarity between the Cerro Azul Formation and Pleistocene–Holocene deposits (Fig. 3). MDS plots depict samples in cartesian coordinates, with similar samples plotted in close proximity and dissimilar samples plotted farther apart[59]. In Fig. 3, samples from the Cerro Azul Formation are plotted alongside samples collected from modern rivers, surficial sediment, and bedrock samples in the Andes. Cerro Azul Formation samples overlap with or are adjacent to the Pleistocene–Holocene samples (Fig. 3). Importantly, the MDS plots show the greatest similarity between the Cerro Azul Formation U-Pb data and that of the Colorado, Negro, and Desaguadero rivers, which are known to be important nodes in the dust production pathway of aeolian Pleistocene–Holocene strata of the Pampas[39,40]. Of the three regional river systems mentioned, Cerro Azul Formation samples plot closer to data from the Colorado and Negro rivers compared to the Desaguadero River. This suggests deflation from megafan deposits of these southern rivers played a relatively larger role in supplying sediment[42], which makes sense as there is a paucity of compelling evidence for a pre-Quaternary Desaguadero River, although it has been speculated based on the tectonic partitioning of the Andean foreland during late Miocene (e.g., ref. 39). When 0–5 Ma grains in the Pleistocene–Holocene deposits are excluded from the MDS analysis, the Cerro Azul Formation samples plot even more closely to the Pleistocene–Holocene samples in MDS space (Fig. S2).

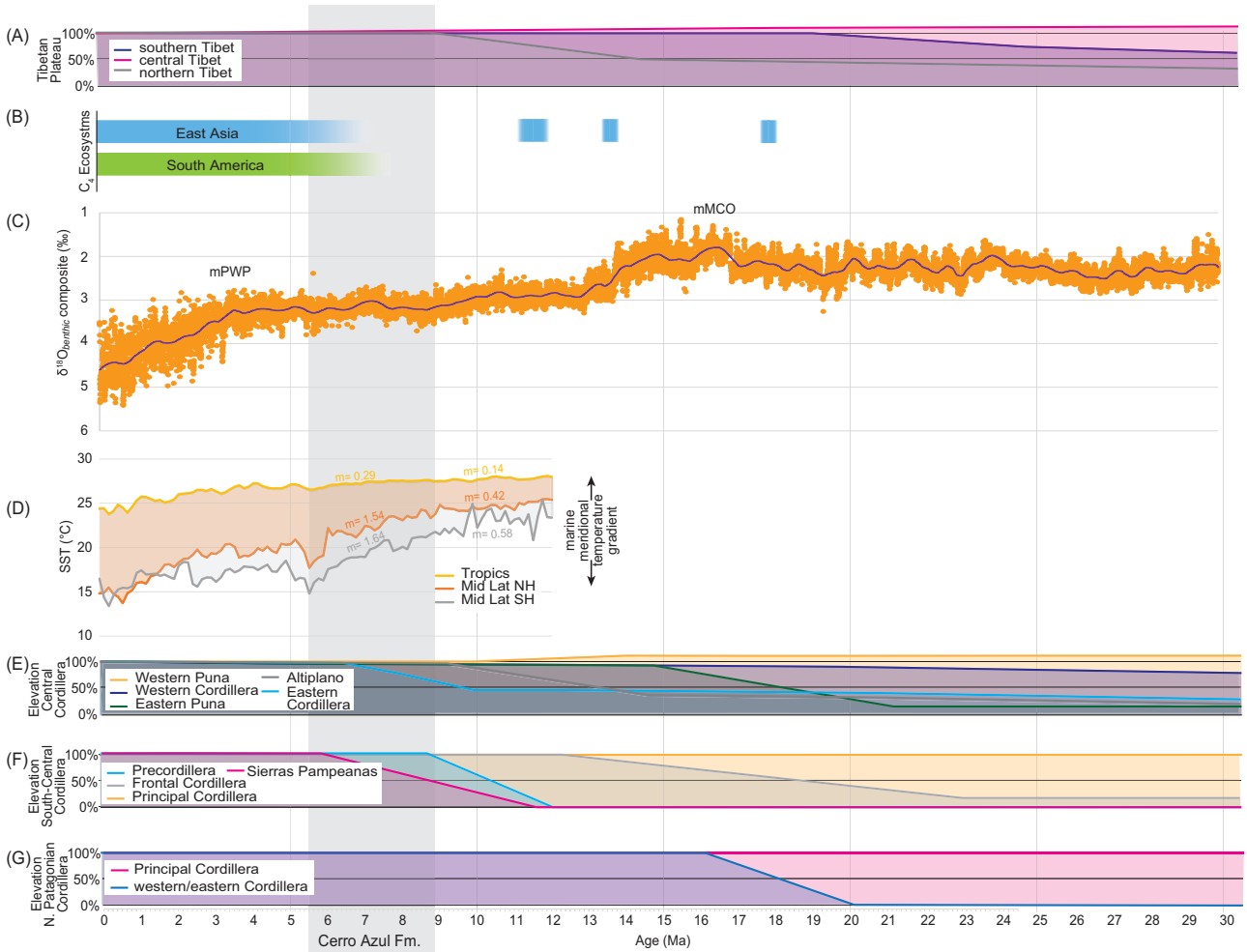

**Fig. 4 | Summary of uplift, C₄ ecosystems, benthic foraminifera δ¹⁸O, and sea surface temperature estimates. A** A synthesis of elevation changes across the Tibetan Plateau where the present-day elevation is 100%. **B** C isotope-based construction of C₄ ecosystems. **C** Benthic foraminifera δ¹⁸O (‰ Pee Dee Belemnite) composite record from ref. 93 showing the mid-Piacenzian Warm Period (mPWP) and mid Miocene Climate Optimum (mMCO). **D** Stacked $U^{K'}_{37}$ sea surface temperature estimates from ref. 26. These include the Tropics, mid latitude Northern Hemisphere (Mid Lat NH; 30°–50°N), and mid latitude Southern Hemisphere (Mid Lat SH; 30°–50°N). Slope (m) calculated from linear regression through 5.50–8.75 Ma (i.e., the interval overlapping deposition of the Cerro Azul Formation) and 9.00–12.25 Ma. **E–G**) Elevation changes across the South American Cordillera including the Central Cordillera (-14°–27°S), South-Central Cordillera (-27°–34°S), and Northern Patagonia (-34°–48°S). Sources in Table S3.

The data provide evidence for the late Miocene establishment of a fluvial-aeolian system on the Pampas nearly identical in terms of provenance and sediment transport pathways to the late Pleistocene–Holocene system. Detailed petrologic and U-Pb detrital zircon data from the more recent aeolian strata exposed across the Pampas indicate sediment was transported from the Cordillera as well as uplifted foreland blocks (e.g., Sierras Pampeanas, San Rafael) through the Colorado, Negro, and Desaguadero river systems (Fig. 1) and then deflated from floodplain settings by lower-level westerly and southwesterly winds[38–40]. When discounting the volcanogenic component that is younger than the depositional age of the Cerro Azul Formation, the similarity between the provenance of Pleistocene–Holocene aeolian deposits and the Cerro Azul Formation (Figs. 2, 3) suggests the sediment sources areas and transport pathways have remained consistent and stable between the late Miocene and Holocene. Recycling of some sediment from the Cerro Azul Formation into Pliocene and Pleistocene–Holocene deposits may have occurred, although this is inferred to have been limited owing to the erosion-resistant calcrete beds capping the Cerro Azul Formation[30]. The volume of Pliocene and Pleistocene–Holocene sedimentary deposits in the Pampas requires a prolonged influx of sediment to the

foreland. We propose that this long-lived fluvial-aeolian system is the product of Andean tectonics, which provides a continuous influx of sediment and influence on lower-level wind patterns, combined with more arid (or seasonal) regional conditions that emerged during the late Miocene[27,48]. This system may have operated largely uninterrupted since the late Miocene; alternatively, the fluvial-aeolian system may have been suppressed (e.g., supply-limited) or altered during the Pliocene to the early Pleistocene as the foreland depocenter migrated south and east surrounding the Pampa Central block, which had become a structural high[42].

Orographic blocking of westerly derived moisture sources by the South American Cordillera was prominently in place before deposition of the Cerro Azul Formation began (Fig. 4). During the late Miocene moisture in the Pampas was derived primarily from the north and east[48,60]. This points to a forcing agent beyond late Miocene orographic development to explain the aeolian deposition within the Cerro Azul Formation. As such, we now assess potential climatic drivers of southern South American dust dynamics in the late Miocene. Meridional sea surface temperature gradients between the tropics and mid-latitudes of the Northern and Southern hemispheres increased dramatically during the late Miocene, and thus point to a concomitant

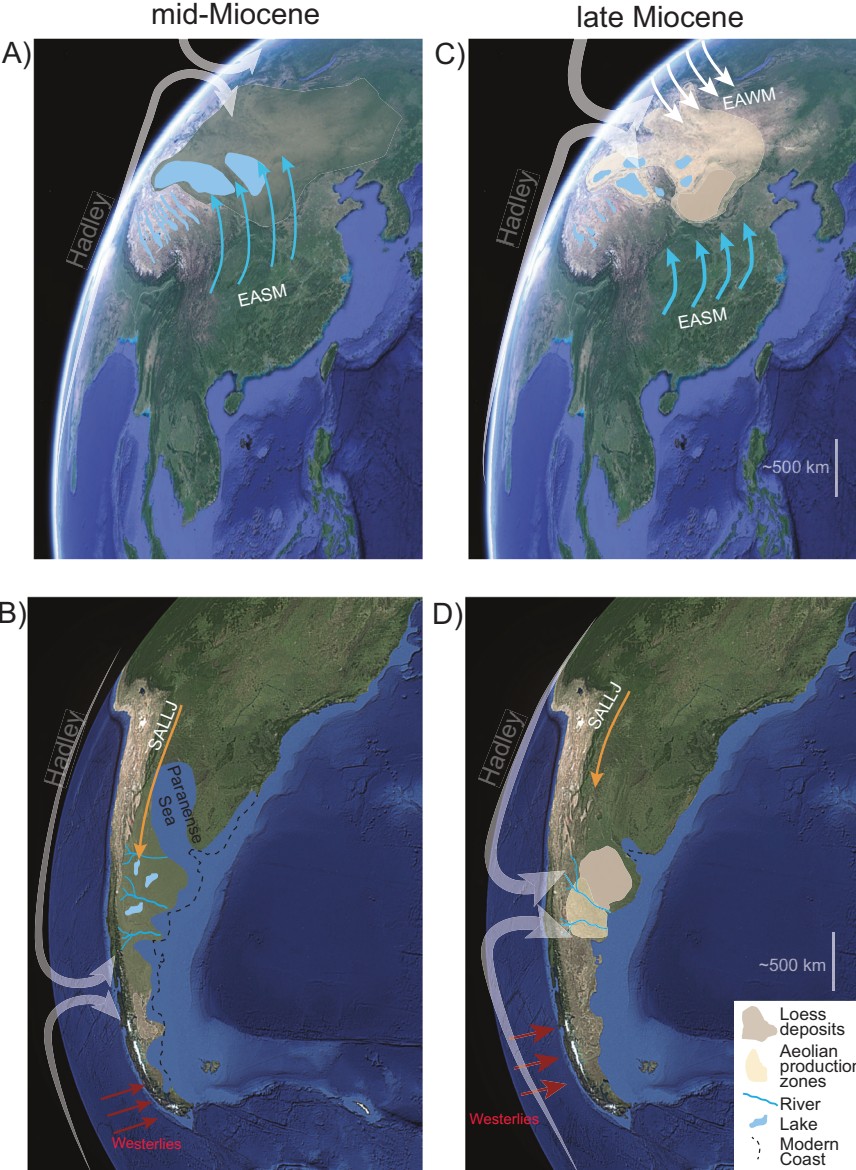

**Fig. 5 | Simplified model of Hadley circulation in the Northern and Southern Hemisphere during the middle Miocene (ca. 15 Ma) and late Miocene (ca. 7–8 Ma).** Schematic distribution of lakes (light blue polygons), aeolian systems (light yellow polygons); and loessoid deposits (brown polygons). **A** Weak Hadley circulation extending into the East Asian continental interior; diffuse extended penetration of East Asian Summer Monsoon (EASM) precipitation[94,95]. **B** Weak Hadley circulation extends south past the Pampas region. **C** Intensification and contraction of Hadley circulation, enhanced drying of the Asian continental interior[70], incursion of the East Asia Winter Monsoon EAWM;[96], and intensification and diminished penetration of the East Asian Summer Monsoon[97]. **D** Intensification and contraction of Hadley circulation[48] and establishment of the Pampas aeolian system. SALLJ, South American Low-Level Jet. The base images are from Google Earth Pro using Landsat: Copernicus.

climate shift at the synoptic scale (Fig. 4D)[26]; Changes in Hadley circulation are one possibility, as there is a strong association between sea surface temperatures and the latitudinal extent and intensity of Hadley circulation (e.g., refs. 61–63). Specifically, previous work posited that the intensification and contraction of Hadley circulation limited the export of moisture from the Amazon basin to the Pampas, as evidenced in central and southern South America through carbon and oxygen isotope data in pedogenic carbonates and fossil enamel, and fossil crown height[48,64]. We suggest that this contraction of the Southern Hemisphere Hadley cell would have ultimately allowed for the establishment of the fluvial-aeolian system in the southwestern Pampas during the late Miocene (Fig. 5). In particular, this increased aridity (or seasonality) in the Central Andes, Sierras Pampeanas, and Pampas (Fig. 1) altered vegetation coverage and increased the

likelihood of aeolian-dominant transport in the Andean foreland basin[15,29]. Paleoclimate models support our assertion, as they demonstrate more arid conditions in the late Miocene Pampas and Sierras Pampeanas would have resulted from changes to synoptic-scale atmospheric circulation patterns that reduced moisture export from tropical South America[48].

Once established in the late Miocene, the fluvial-aeolian system of the Pampas has undergone only moderate changes to sediment source areas and routing as demonstrated by the similarity in provenance between upper Miocene and Pleistocene–Holocene aeolian deposits (Fig. 2). One potential modification is the establishment of the paleo-Desaguadero River. Additionally, the detrital zircon U-Pb data indicates an increased late Miocene–Pleistocene volcanogenic sediment load over the last 5 million years, either through rapid recycling or

direct input[40]. Regardless, this consistency has occurred despite dramatically shifting climate, exemplified by both prolonged cooling from the late Miocene through the Quaternary as well as the approximately 3-Myr warmer-than-present conditions encompassed within the Pliocene Epoch (Fig. 4B–D).

Regarding the Pliocene, the late Miocene fluvial-aeolian system likely persisted in some form across the Pampas through this interval, as evidenced by Pliocene-age aeolian deposits preserved in the Colorado Basin of eastern Argentina (Fig. 1)[42]. However, in this scenario, the Pliocene to middle Pleistocene Pampas was an area of net sediment bypass or erosion, where deposited sediments were ablated from the landscape prior to accumulation of upper Pleistocene–Holocene strata that unconformably overlie the upper Miocene Cerro Azul Formation. Windblown dust deposits (i.e., loess) have low long-term preservation potential in the geologic record with as much as 80% of loess being eroded or reworked before preservation can occur[65]. We speculate that the 1–2 m thick calcrete duricrust that caps the Cerro Azul Formation[30] delimited erosion of any overlying Pliocene and lower Pleistocene aeolian strata that accumulated across the southwestern and central Pampas. The resistant calcrete bed above the Cerro Azul Formation prevented the erosion of the upper Miocene strata—thus, becoming the exception rather than the rule in preservation. Further, we hypothesize that the documented increased aridity commencing in the late Miocene made the Pliocene southwestern Pampas more supply limited than the late Miocene system until the southward development (or rerouting) of the paleo-Desaguadero River system. An alternative to this explanation suggests that the Pampas fluvial-aeolian system was supply-limited at the Miocene-Pliocene transition until the re-expansion and weakening of the Hadley cell during the early Pliocene (Fig. 5)[26,66] drove intensified precipitation and sediment accumulation that was still outpaced by wind erosion in many areas. Although a large portion of the late Pleistocene–Holocene Pampean loess is composed of sediment recently deflated from the Desaguadero, Colorado, and Negro river systems as evidenced by petrology and detrital zircon U-Pb data[39,40], a component of the detritus composing the late Pleistocene–Holocene strata could likely have been recycled from the upper Miocene Cerro Azul Formation. This is supported by the up to 100 m of relief in valleys within the uplifted and tilted Pampa Central block, thought to be the result of dominant aeolian erosional processes, supplemented by fluvial erosion[67]. Notably, the WSW-ENE trending valleys form a continuum with the smaller-scale linear deflation features of the Central Pampean dune field[13]. Along with the dearth of Pliocene–lower-middle Pleistocene strata in the southwestern and central Pampas, this aeolian continuum implicates wind erosion down to the calcrete horizon that overlies upper Miocene strata.

To put the provenance of the upper Cerro Azul Formation and the establishment of a late Miocene–Holocene fluvial-aeolian system in central South America in a global context, we draw comparisons with the Chinese Loess Plateau (Fig. 5). Although the aeolian Chinese Loess Plateau record extends into the early Miocene[68] or earlier[69], these loessoid deposits are relatively limited compared to the spatial extent of the late Miocene–Quaternary Chinese Loess Plateau[70]. The extensive loessic paleosol deposits of the red-clay sequence of the Chinese Loess Plateau, exposed beneath the Quaternary loess and paleosol sequences, started accumulating at ca. 8 Ma[71–73]. Like the Andean-Pampean scenario, most uplift of the Tibetan Plateau occurred prior to the accumulation of the red-clay sequence of the Chinese Loess Plateau. This implies the establishment of both the Pampean and the Chinese Loess Plateau aeolian systems began in earnest during late Miocene cooling at ca. 8 Ma. Although influenced by much different precipitation systems—the Chinese Loess Plateau is located at the limit of East Asian Summer Monsoon penetration into Asia and is influenced by moisture from within the Asian interior[74], whereas the Pampas are influenced by the South American lower-level jet and atmospheric

convection in the Amazon (Fig. 5)—and with different orographic frameworks with respect to moisture sources, both systems occur between ~33°–39° latitude in their respective hemispheres and have similar present-day spatial footprints (1–5*10^5 km²). Combined, these observations suggest a global process behind the coeval establishment of these systems. Building on our interpretation for the Cerro Azul Formation and previous work, we posit that bihemispheric intensification and contraction of Hadley circulation in response to late Miocene cooling largely forced the establishment of these two spatially extensive and long-lived aeolian provinces (Figs. 4D, 5)[26,48,75]. Sufficient intensification and equatorward contraction in the Northern and Southern Hemispheric Hadley cells, likely related to changes in meridional temperature gradients, would have needed to occur between the mid-Miocene Climate Optimum (ca. 15 Ma) and ca. 8 Ma to place these latitudes near the descending atmospheric limb[26,66]. A prominent increase in meridional sea surface temperature gradients in the late Miocene (ca. 7.5–5.5 Ma) fits this hypothesis (Fig. 4D)[26]. We note that both aeolian systems are primarily defined by aggradational loessic paleosols that started accumulating in the late Miocene, which implies similar humidity, temperature, landform stability, and possibly vegetation (Fig. 4B) at that time. Furthermore, this hypothesis does not rely on Tibetan Plateau uplift as a main forcing agent in the expansion of the Chinese Loess Plateau around 8 Ma. Parallels between the onset of aeolian sedimentation in the Chinese Loess Plateau and the Pampean aeolian system suggest increasing meridional temperature gradients (and their associated drivers) exerted a prominent influence on mid-latitude aridification during the late Miocene.

## Methods

### Sampling and mineral separation

We collected $N = 9$ samples of loess and loessic paleosol from exposures of the upper half of the Cerro Azul Formation located along the southwestern margin of the Pampas (Fig. 1). Three samples were collected from the type section of the Cerro Azul Formation near the Rio Desaguadero[30]; the others were collected from outcrops across a ~300 km east-west transect (Fig. 1). Measured sections were completed at sampled outcrops. Samples were processed for detrital-zircon recovery using density and magnetic mineral separations. Detrital zircon grains with sizes of silt to very-fine sand were extracted from samples using standard separating procedures[76]. All disaggregated sample material was passed through a 500-μm sieve and <500 μm material was further separated using a Gemeni water shaker table. Grains were separated using a Frantz barrier field isodynamic magnetic separator and differentiated using lithium metatungstate heavy liquid. Zircon grains were mounted in 2.5 cm diameter epoxy and polished with 2000 grit sandpaper to expose grain cores.

### Laser-ablation inductively-coupled-plasma mass-spectrometry

U-Pb ages of individual grains were measured using laser-ablation inductively coupled plasma mass-spectrometry (LA-ICP-MS) at the Arizona LaserChron Center of the University of Arizona, yielding a total of $n = 3299$ U-Pb ages with a range of $n = 198–541$ per sample, allowing some robust comparison between the relative proportions of ages within age modes[77]. Laser spot size used for analyses was 20 μm. Elemental and mass-fractionation instrument drift and down-pit fractionation were corrected using a suite of reference materials mounted with the zircon samples. The reference material included Sri Lanka ('SL'), FC-1, and R33[76]. The corrections were made using an in-house, Microsoft Excel-based software (AgeCalc). The $^{206}Pb/^{238}U$ versus $^{207}Pb/^{206}Pb$ age cut-off used was 900 Ma thus zircon crystal ages with $^{206}Pb/^{238}U$ ages <900 Ma were calculated using the $^{206}Pb/^{238}U$ ratio and crystal ages with $^{206}Pb/^{238}U$ ages >900 Ma calculated using the $^{207}Pb/^{206}Pb$ ratio. We employed the following rejection criteria: [1] maximum $^{204}Pb$ signal intensity of 100 cps; [2] maximum $^{206}Pb/^{238}U$ uncertainty of 10%; [3] maximum $^{207}Pb/^{206}Pb$ uncertainty of 10%; [4]

maximum $^{206}Pb/^{238}U$ vs. $^{207}Pb/^{206}Pb$ age discordance of 30%; or [5] maximum $^{207}Pb/^{206}Pb$ vs. $^{206}Pb/^{238}U$ reverse age discordance of 10%.

## Data availability

The U-Pb data generated in this study have been deposited in the Figshare database (https://doi.org/10.6084/m9.figshare.23398742); the minimum dataset necessary to interpret, verify, and extend the research in the article is available in the supplementary material and through Figshare.

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

## Acknowledgements

We thank F.J. Prevosti and A. Forasiepi for logistical support and L. Tully for assistance with sample preparation and processing. The research reported here was performed by an international team of geoscientists including researchers native to the study areas. All roles and responsibilities were agreed upon amongst the collaborators and freely communicated within the team. Previous local and regional research was fundamental to the success of our research and widely incorporated into this report. This research was funded by the U.S. National Science Foundation: U.S. National Science Foundation grant EAR 1910510 (A.L.L. and D.L.B); U.S. National Science Foundation grant EAR 1911340 (A.P. and M.K.F.); U.S. National Science Foundation grant EAR 1545859 (C.N. Garzione; subaward to A.P.); J.T.A. is currently supported by the U.S. National Science Foundation (NSF-OCE-PRF #2126500).

## Author contributions

A.L.L., D.L.B., A.P., and M.K.F. designed the project. B.S., D.L.B., and A.P. completed the field investigation and sample collection. B.S. was responsible for sample preparation. B.S., A.L.L, D.L.B., and A.P. were responsible for data collection and synthesis. B.S., A.L.L., D.L.B., A.P., M.K.F., J.T.A., J.N., and M.A.Z. made intellectual contributions to and wrote the manuscript, and all authors contributed to comments and revisions.

## Competing interests

The authors declare no competing interests.
