## [Peer Review File · Nature Communications]

Global climate forcing on the late Miocene establishment of the Pampean aeolian system in South AmericaREVIEWER COMMENTS

Reviewer #1 (Remarks to the Author):

In this paper, Stubbins et al. document a potentially long (since Miocene) record of an aeolian-fluvial system in South America. These types of long records are very rare, and so this is quite noteworthy. Based on their convincing age and provenance data, aeolian activity continued (perhaps not uninterrupted) from the Miocene into the Pleistocene-Holocene. This is based on their data showing that the provenance of loess from Miocene and Pleistocene deposits has remained unchanged aside from more recent volcanic contributions to the Pleistocene loess. Rivers draining the Andes are a key source of sediment for this long-lived aeolian system and connects the importance of the tectonic development of the Andes to this aeolian system. The story is straightforward and significant by highlighting the importance of dust in the Southern Hemisphere and linking tectonics, fluvial and aeolian systems, and climate change, increasing its relevance to a broad audience. The importance of aeolian-fluvial systems is being recognized more frequently in recent research, and this will be an important contribution to that work.

The paper makes an interesting comparison to the Chinese Loess Plateau, suggesting the two are linked by global climate change and Hadley cell circulation patterns. I find this link exciting and compelling. The writing is very good and the logic is easy to follow, aside from a few long sentences. I found the figures generally easy to read and interpret (I had to study Fig 1 to see where the Cerro Azul Fm was located, as the dotted pattern doesn't come through very well in the key), and the supplemental figures were useful to show the strong similarities in provenance between the Miocene and Pleistocene-Holocene loess. I recommend publication with minor edits.

62: Did aridity and seasonality directly increase the sediment production and deposition? The sentence skips some steps, so to soften that, I suggest the following phrasing: aridity and seasonality is linked to increased sediment production and deposition.

163-168. Long sentence. I suggest breaking this up into two or revising.

190-194: Another long sentence. If kept intact, replace allowing with allowed. And perhaps replace the dashes with commas.

Reviewer #2 (Remarks to the Author):

Summary:

This is a very interesting paper that uses detrital zircons to explore an important question about the rise of continental-scale aeolian systems. Their main finding is that pooled DZ spectra from the Miocene Cerro Azul Formation match well with the pooled DZ spectra from the aeolian sands and loss on the modern Argentine Pampas. They then build on this similarity to propose that the aeolian system was initiated in the Miocene and has persisted until today. Ultimately they extend their findings to argue for a global forcing mechanism to establish large mid latitude loess provinces.

Although these conclusions are interesting and plausible, the potential impact of the 'big ideas' is diminished by their relatively unconvincing arguments for the persistence of the Pampean aeolian system. For example, the core finding of the paper is that the pooled DZ spectra are similar between the Miocene Cerro Azul Fm. and the modern Pampean sand and loess. They note that the pooled spectra are quite different from other regional sources such as the Puna and Patagonia. To me, this seems like a very intuitive finding given that the sand and loess is found down wind of the Pampean river systems that generate it. The key conclusions I would draw from this dataset are that the pooled DZ spectra are consistent with the geology of the Pampean source region and have been blown to the east, likely recycled many times as the climate has oscillated between drier and wetter conditions. It is not clear whether the aeolian system has truly persisted since the Late Miocene, or whether zircon crystals themselves have persisted as they were intermittently pushed eastward. To me this is an important distinction that can only be resolved by more careful stratigraphic synthesis, demonstrating that there are actually eolian deposits that span the entire time interval between Miocene and present. I suggest that the authors could do more to demonstrate that the source regions are truly identical between the Miocene Cerro Azul and Pleistocene samples. I worry that pooling the DZ spectra masks some complexity that could point to a more complex story. I propose an additional figure showing the individual spectra as CDF (or KDE). It would also be a good idea to include all of the DZ analyses in a supplemental data table for use by future researchers.

In sum, my main comment is that the paper should be reframed a bit and presented in a more nuanced way to directly address some of the major assumptions that underly the more speculative conclusions.

Specific comments:

Lines 54-55: My intuition is that relatively little of the Andean foreland sediment is late Pleistocene in age. That said, please clarify your thinking. Are you talking about foreland basin sediments or the sand veneer covering the more distal Pampas?

Lines 60-62: This comment makes it sound like the late Miocene cool conditions continued directly into the latest Pliocene/Pleistocene. Don't forget about the Pliocene warm period, which likely marked a return to warm temps between the late Miocene and Pleistocene. I suggest bringing in a bit more detailed background info about what is known about the Pliocene terrestrial climate of Argentina here. Admittedly, this may not be much...

Lines 68-70: To me it is intuitive that this system has been in place in some form since the rise of the Andes. Rivers bring sediment to the foreland and it is blown west. I think the paper would benefit from a more nuanced analysis of this question that includes a broader stratigraphic synthesis. Are there aeolian deposits older than the Cerro Azul Fm.? Are there aeolian deposits during the Pliocene to signal the system persisted? I recognize that later in the paper the authors discuss the relative lack of Pliocene deposits, suggesting they have been largely deflated.

Lines 81-84: This seems like the key point to me.

Lines 90-92: Is this an alternative 'straw hypothesis'. Consider developing it further and clarifying it. If this straw hypothesis is correct, would it predict something very different from your findings?

Lines 109-112: I feel it is important to present the DZ spectra from all samples individually, without pooling them. At least as a supplementary figure. Although the MDS technique is exciting, to me it is not a replacement for presentation of the full DZ spectra. This would allow the results to be visualized without the overlay of a fairly complex and opaque statistical technique. Individual spectra could be plotted either as KDE or CDF. The CDF might be more appropriate as it directly relates to the KS statistic that underpins the MDS technique.

Lines 139-141: Can you offer more detail on what method you are using to estimate this similarity coefficient? Also, can you give us a sense of how similar the samples within each of the pooled groups are (i.e. within the Cerro Azul and Pleistocene groups).

Lines 159 to 160: It could be helpful to explain how the other regional samples were obtained and estimated. Are they from pooled river samples? Pooled bedrock samples? Is there potential for grain size biasing between these depositional environments and an aeolian environment?

Lines 173-174: Here I think it is important to clarify that you show evidence for 'nearly identical' aeolian zircon spectra. Not sure it is justified to jump to the conclusion that the systems were nearly identical. Seems that would require a more details stratigraphic analysis.

Lines 181-183: Similar comments to above. It is not clear to me that the Pleistocene samples were transported from their original bedrock source during the Pleistocene, but instead are likely reworked over many climate cycles.

Lines 204-206: This is an important point to strengthen your arguments. Can it be further developed? Are their Pliocene aeolian deposits elsewhere in the study area?

Lines 211-213: Clarify what you mean by 'delimited' here. Are you essentially saying that the overlying Pliocene aeolian sediments were eroded down to the duricrust, which prevent deeper erosion?

Lines 213-220: Great that you are addressing what might have happened during the Pliocene. Seems this uncertainty undermines the strength of the argument for a single identical system persisting from Miocene to Pleistocene.

Lines 266-269: What grain size was ultimately analyzed? Was further sieving performed prior to mounting/analysis? Is there any concern over grain sized biasing when comparing loess-derived zircons to fluvial systems, etc?

Figure 1: Can you clarify what is meant by loess samples? Those are only Cerro Azul samples, correct?

Figure 2: Would still like to see individual KDEs or CDFs for each sample. Consider labeling y-axis as “# of grains” or similar.

Figure 3: At some point it would be great to unpack how these other comparison spectra for SP, Chaco, Patagonia, and Puna were compiled and what geologic context they are from

Figure 4: might label ‘westerlies’ in red text to match arrows.

Reviewer #3 (Remarks to the Author):

In this study, Stubbins et al. present new U-Pb ages of detrital zircons, largely of late Miocene age, from the Argentinian Pampas region. Their data show that the provenance of these grains is very similar to those found in Pleistocene and Holocene sediments and hence that the fluvial-aeolian processes responsible for the Pampas system have been in operation since the late Miocene. They argue that this result demonstrates that global cooling was responsible for the generation of large mid-latitude loess provinces in the late Miocene.

The conclusion that there has been little change in the provenance of sediments within the Pampas since the late Miocene is generally well supported by the new data presented here with a clear and transparent methodology. However, I find that “global climate forcing on the late Miocene establishment of the Pampean system” (as stated in the title) is not currently sufficiently evidenced within the manuscript to support this as the main conclusion of this study. Most of the evidence for this second conclusion seems to come from previous work, and very little of this is shown within this paper (although relevant studies are cited and their conclusions stated in the text). In addition, several of the cited references already make the link between late Miocene aridification and regional sedimentation (e.g. 27-29), so I think a clearer statement of how this study represents a significant advance on what is already known and/or tackles an outstanding question would be beneficial.

In order to prove that global climate forcing was responsible for the establishment of the Pampean system, I think there are four key points that need to be clearly demonstrated: 1) The Pampas was established in the late Miocene. 2) A dominant local tectonic driver of this event can be ruled out. 3) There was local climatic change at this time that can explain the establishment of the Pampean system. 4) This climatic change can be linked to/explained by global factors. Regarding each of these four points: 1) There are some new ages presented in this paper, which are reported to be consistent with previous work. If these new ages are adding additional value, this could be more clearly emphasised to help strengthen this manuscript. Also, as the timing of loess formation is key to the main conclusions, a figure or panel demonstrating the distribution of reported ages would be helpful. The late Miocene spans ca. 6 million years. Can you be any more precise about the likely timing of Pampas establishment and/or comment on whether it was rapid or gradational? 2) The text currently states that the “fluvial-aeolian system is the product of Andean tectonics, which

provides a steady influx of sediment and influence lower-level patterns, combined with more arid conditions that emerged during the late Miocene". Can you comment on how important these two drivers are relative to each other? This is crucial to how strongly the message that global climate was the main driver can be stated but is not currently discussed in detail.

3) L190-196 list a range of previously published evidence for regional climate change. This is really important to the key message, and I think the paper would benefit from plotting some of these data (or a summary of their results) in a figure to allow readers to see this evidence for themselves. This could be combined with the age data in point 1.

4) Links are made in the text to both late Miocene cooling and the development of the Chinese Loess Plateau in the text, but again, no evidence is directly presented to the reader within this study to demonstrate how similar timings/patterns of change are to justify the strength with which this key message is presented. This could also be shown as part of a figure.

In summary, I find that this study presents new provenance data that clearly demonstrate the relatively stability over millions of years of a major fluvial-aeolian system, and has the potential to be a paper worthy of publication in Nat. Comms. However, in my opinion, the main conclusion of the paper is not currently well enough supported to justify publication in the current form. I therefore suggest that either the authors provide more evidence to support their conclusions so that their key message is well supported without requiring further reading (my recommended option to produce a manuscript of interest to a broad audience), or they change the manuscript title and reduce the confidence with which they link their records to climatic drivers.

Line-by-line comments

L31: "a global forcing mechanism" is quite vague.

L35-36: This implies that there was no wind-blown dust before the establishment of the Pampas, which seems unrealistic!

L44: Which instances?

L44-50: There's a bit of a reasoning jump here. Why is S. America in particular "critically important"?

L58, L64: "western Pampas" v "southwestern Pampas" – is the distinction significant? As a general point, please make figure 1 tie into the text more closely so that a reader unfamiliar with the study area can easily identify the features described.

L64-65: Reference?

L64-L72: It's not clear what's being stated as already known and what's presented as the novel findings of this study in this paragraph.

L71: “similar mechanism” rather than “thesis”.

L90-92: Phrase this as a hypothesis? Are you trying to set up the idea that large province shifts might still be expected in the latest Miocene and that’s something that your data will allow you to explore?

L91: Define bajadas

L93-106: Some repetition here with L58-72.

L103-L104: More arid/greater aridity and/or seasonality than what?

L108-134: More explanation would be useful throughout the results section to help a non-specialist audience.

L109-110: A comment on the geographical spread of the samples would be useful here.

L110-111: What is/is the purpose of a KDE?

L116: What about the lithology of lower half of the Cerro Azul formation? Is this relevant in the aridification history of the region?

L117: Explain “U-Pb zircon age modes”.

L138-139: From the same locations? Or averaged across region/exposures?

L143: “the same” rather than “equivalent”?

L145: “all samples” stated earlier – do some Pleistocene samples differ?

L159-162: The phrasing of this sentence is quite confusing

L163-168: What determines which sources are the most important? Wind/river direction? Erodibility? Extent of exposure? Other?

L173-178: This is an example of a section where clearer integration with Fig 1 would be helpful.

L183-185: How important are these two factors? References needed for late Miocene aridification.

L187: Unclear which part of the sentence “possibly” refers to.

L185-189: How do deposition rates change?

L204-206: Confusing – “may have” and “it did so” seem to contradict each other. Are you saying it persisted in one location but not extensively?

L218: Any other lines of evidence to support this change in Hadley cell dynamics?

L237: Three of your 9 samples have MLA >9Ma, so can the formation of the Pampas be directly linked to the 8Ma start of the CLP? This age discrepancy should be acknowledged and discussed.

L244: Why is the similarity in spatial footprints relevant?

L249: “would be needed” instead of “had to occur”.

L255-257: “the global nature of our findings” currently not supported by the presented data

L267: Clarify if >500 or <500µm fraction kept.

L292-293: Part of sentence missing?

L314: “Simplified” not “general” model to avoid any confusion with GCMs?

L324: Please carefully check reference formatting e.g. L350

L686: Figshare link not currently working, but I assume would be made live upon acceptance?

Figure 1: I find this figure messy and overly complex. I suggest aligning it more closely with the aspects of local geography discussed in the text to help the reader follow the main arguments. Is it possible for the geological units shown to more closely align with the 8 age modes used in the study? What does all of the white space represent? Is the Cerro Azul represented by dots on both white and yellow backgrounds? Figure would look smarter if framed by a box. Also ensure latitude tick marks are outside edge of map (40S appears to be within national boundary).

Figure 2: Add y-axis labels & ensure tick marks touch axis.

Figure 3: Label axes. Need to acknowledge source of background image in panel B?

Figure 4: Add “aeolian production areas” to legend. Can’t see lossoid deposits anywhere on the map? What is the significance of the different arrow colours? Define SALLJ.

S1: Ensure CaCO₃ formatted correctly throughout. What is “Type 1 Locality”?

Authors Responses to Reviewers' comments

(AR = Authors' Response; line #s refer to "clean" manuscript, sans "track changes")

Reviewer 1:

62: Did aridity and seasonality directly increase the sediment production and deposition? The sentence skips some steps, so to soften that, I suggest the following phrasing: aridity and seasonality is linked to increased sediment production and deposition.

AR: we have rephrased, as suggested. Line 65-66

163-168. Long sentence. I suggest breaking this up into two or revising.

AR: we have broken this into two sentences. Line 175

190-194: Another long sentence. If kept intact, replace allowing with allowed. And perhaps replace the dashes with commas.

AR: we have changed the sentence following the reviewer's advice. Line 218

Reviewer 2:

General comments:

a) It is not clear whether the aeolian system has truly persisted since the Late Miocene, or whether zircon crystals themselves have persisted as they were intermittently pushed eastward. To me this is an important distinction that can only be resolved by more careful stratigraphic synthesis, demonstrating that there are actually eolian deposits that span the entire time interval between Miocene and present.

AR: The reviewer raises an interesting point that requires a bit of clarification. The reviewer is asking the hypothetical, or alternative, hypothesis: What if the sediment was initially delivered to the foreland basin during the Miocene and simply been recycled over the last ~6 million years?

As the reviewer mentions, this is an important consideration. We agree that some sediments may have been recycled over time, particularly amongst the sand-sized fraction. However, our samples consist of loess and loessoid deposits, and it is highly unlikely loess would be continuously recycled within the same basin over the past ~6 million years. There are 4 considerations that do not support the 'recycling' hypothesis:

1) There is not enough material in the Cerro Azul Formation to account for the vast Pleistocene-Holocene deposits (and Pliocene deposits as well). This necessitates a continued influx of sediment from the Andes to the foreland since the Late Miocene.

We added a statement to the text to clarify this: Line 195-197

2) The ~2 m calcrete horizon that overlies the Cerro Azul Formation serves as an "erosional cap," preventing extensive erosion of the Cerro Azul Formation. This means that overlying eolian sediment in the basin must have been derived from coeval foreland river systems and not recycled. This is discussed in Line 238-251

3) Although not stated explicitly, implicit within Line 245, where the low preservation potential of loess is discussed, lies the notion that recycling the same loess-sized material (silt) within the same basin is highly unlikely.

4) Pliocene eolian deposits exist in central Argentina east of our study area, supporting a relatively continuous aeolian system between Miocene-Holocene – in contrast to a recycled hypothesis. Lines 239-240.

Two other issues are tangentially raised in this discussion, which we have addressed. When discussing this fluvial-aeolian system, we are not only describing the aeolian system, but also the requisite components of the system, which include the sediment source (various ranges within the Andes), basin, foreland rivers, and atmospheric-climate conditions conducive to eolian transport. The near-identical detrital zircon record from late Miocene eolian deposits and Holocene eolian deposits makes it clear that this 'system' was in place during the late Miocene and has been operating since that time (i.e., persisted). To clarify this, we have made various changes (see specific details below) to better define this in the text, including: "...The fluvial-aeolian framework that has directed dust production, transport, and deposition in the Pampas consists of several components including sediment sources, foreland rivers and climate/atmospheric conditions..." Line 72-73

With respect to the general comments - A (Part 1):

We agree with the reviewer that this does not necessarily mean that the aeolian system was active throughout that entire duration – certainly climate cycles and other variables have impacted aeolian activity in the Pampean aeolian system over time. This is mentioned several times in the text as well – e.g., Line 246-255

With respect to the general comments B:

b) I suggest that the authors could do more to demonstrate that the source regions are truly identical between the Miocene Cerro Azul and Pleistocene samples. I worry that pooling the DZ spectra masks some complexity that could point to a more complex story. I propose an additional figure showing the individual spectra as CDF (or KDE). It would also be a good idea to include all of the DZ analyses in a supplemental data table for use by future researchers.

The Reviewer brings up a very good point regarding the data included in this manuscript. The pooling of ages into Miocene and Pleistocene samples is done intentionally in order to make 1st order comparisons between the two aeolian systems (Miocene versus Pleistocene). In other words, the focus of this study is to make the broader interpretations, which is most easily done by pooling ages. However, pooling data can obfuscate the individual sample results to some extent. In the case of this, study, this is not a concern – the individual Miocene samples display little to no variation and are consistent with the pooled ages. Nonetheless, the reviewer is correct in the assertion that we need to have these individual samples available to the reader.

To rectify this, we have created a new figure (FigS1 – Supplementary Figure 1) of all of the individual KDEs from each sample and added this as an additional figure for the supplementary data section.

Specific comments From Reviewer 2:

Lines 54-55: My intuition is that relatively little of the Andean foreland sediment is late Pleistocene in age. That said, please clarify your thinking. Are you talking about foreland basin sediments or the sand veneer covering the more distal Pampas?

AR: yes, we are referring to the material at the surface today. We've altered the text accordingly in order to make this clear. Line 57

Lines 60-62: This comment makes it sounds like the late Miocene cool conditions continued directly into the latest Pliocene/Pleistocene. Don't forget about the Pliocene warm period, which likely marked a return to warm temps between the late Miocene and Pleistocene. I suggest bringing in a bit more detailed background info about what is known about the Pliocene terrestrial climate of Argentina here. Admittedly,

this may not be much...

AR: Because this is the Introduction, we are hesitant to add details that could distract from the key points. However, we discuss this issue later in the text - Lines 235-237

That being said, paleoclimate records indicate an overall cooling since the late Miocene. It is true that the mid-Pliocene warm period (or mid-Piacenzian warm period) is associated with increased temperatures, but this is a relatively minor departure from the overall trend. Moreover, temperatures during mid-Pliocene warming were still cooler than those in the late Miocene (Prevosti et al., 2021; Herbert et al., 2016 and references therein). As the reviewer suggests, information on the Pliocene terrestrial climate of Argentina is sparse, and the Pliocene sedimentary record in our study area is non-existent.

Lines 68-70: To me it is intuitive that this system has been in place in some form since the rise of the Andes. Rivers bring sediment to the foreland and it is blown west. I think the paper would benefit from a more nuanced analysis of this question that includes a broader stratigraphic synthesis. Are there aeolian deposits older than the Cerro Azul Fm.? Are their aeolian deposits during the Pliocene to signal the system persisted? I recognize that later in the paper the authors discuss the relative lack of Pliocene deposits, suggesting they have been largely deflated.

AR: We made changes to the Background section – adding that “Aeolian deposits in the Cerro Azul Formation, recorded as loess-paleosol sequences, represent the oldest aeolian deposits in the Pampean Neogene succession ...” Line 109-110. We did not make additional changes to this specific part of the manuscript (in the Introduction) because we are simply trying to introduce the overall arc of the study.

Lines 81-84: This seems like the key point to me.

AR: No change requested.

Lines 90-92: Is this an alternative ‘straw hypothesis’. Consider developing it further and clarifying it. If this straw hypothesis is correct, would it predict something very different from your findings?

AR: Both Reviewer #2 and #3 had a question about this sentence.

We have redrafted it to make it a clear statement – a statement that summarizes our current understanding of how provenance has changed slightly in response to recent tectonic uplift. This has been demonstrated in Garzanti et al., 2022. The primary point of the sentence (beyond the explicit statement) is to make the reader aware that this is a tectonically active foreland and therefore sediment provenance is constantly evolving as crustal blocks are displaced and eroded. Line 96-98

Lines 109-112: I feel it is important to present the DZ spectra from all samples individually, without pooling them. At least as a supplementary figure. Although the MDS technique is exciting, to me it is not a replacement for presentation of the full DZ spectra. This would allow the results to be visualized without the overlay of a fairly complex and opaque statistical technique. Individual spectra could be plotted either as KDE or CDF. The CDF might be more appropriate as it directly relates to the KS statistic that underpins the MDS technique.

AR: We have added a figure to the supplementary section (Figure S1) as the reviewer requests. We agree that the individual samples should be shown individually.

Lines 139-141: Can you offer more detail on what method you are using to estimate this similarity coefficient? Also, can you give us a sense of how similar the samples within each of the pooled groups are (i.e. within the Cerro Azul and Pleistocene groups).

AR: We have included the reference, which contains the details of how the similarity coefficients were calculated (Saylor and Sundell 2016). Similarity within or between samples is now accessible by observing our newly added figure (Figure S1).

Lines 159 to 160: It could be helpful to explain how the other regional samples were obtained and estimated. Are they from pooled river samples? Pooled bedrock samples? Is there potential for grain size biasing between these depositional environments and an aeolian environment?

AR: we amended the text a bit and added:

“In Figure 3, samples from the Cerro Azul Formation are plotted alongside samples collected from modern rivers, surficial sediment and bedrock samples.” Line 169

Lines 173-174: Here I think it is important to clarify that you show evidence for ‘nearly identical’ aeolian zircon spectra. Not sure it is justified to jump to the conclusion that the systems were nearly identical. Seems that would require a more details stratigraphic analysis.

AR: we have changed the text slightly to address this point. We are not trying to say that the Miocene eolian system is identical to what we see today – e.g., with large eolian dunes (Pampean Sand Sea) and a loess belt. We are trying to say that the source areas and sediment transport pathways are the same.

We added this point to the text to correct this. Line 186

Lines 181-183: Similar comments to above. It is not clear to me that the Pleistocene samples were transported from their original bedrock source during the Pleistocene, but instead are likely reworked over many climate cycles.

AR: This is a good point – and we discuss this at some length in the “general comments” section above. We added that there is some possible recycling. However, there is no way that the Cerro Azul Formation, even if all of it were recycled, could account for the volume of Pliocene and Pleistocene eolian strata in central Argentina. There has to have been a continuous supply of sediment and eolian entrainment and deposition throughout the Miocene-Holocene (see response under general comments above).

We added to the text: “Recycling of some sediment from the Cerro Azul Formation into Pliocene and Pleistocene-Holocene deposits may have occurred, although this is inferred to have been limited owing to the erosion-resistant calcrete beds above the Cerro Azul Formation (cite 43). The volume of Pliocene and Pleistocene-Holocene sedimentary deposits in the Pampas requires a prolonged influx of sediment to the foreland. Line 238-251

Lines 204-206: This is an important point to strengthen your arguments. Can it be further developed? Are their Pliocene aeolian deposits elsewhere in the study area?

AR: There are no Pliocene units preserved in our study area. However, there are aeolian units to the east along the Atlantic margin in a region known as the Colorado Basin. We highlighted this and added the location to the text:

“of eastern Argentina (Fig. 1)” Line 261-262

Lines 211-213: Clarify what you mean by ‘delimited’ here. Are you essentially saying that the overlying Pliocene aeolian sediments were eroded down to the duricrust, which prevent deeper erosion?

AR: Yes, this is correct. There is a ~2 m thick pedogenic carbonate above the Cerro Azul Formation that acts as a “cap” to the unit – it is probably the main reason that the Cerro Azul Formation has not been eroded away.

We added to the text:

“The resistant calcrete bed above the Cerro Azul Formation prevented the erosion of the upper Miocene strata.” Line 248-249

Lines 213-220: Great that you are addressing what might have happened during the Pliocene. Seems this uncertainty undermines the strength of the argument for a single identical system persisting from Miocene to Pleistocene.

AR: Our discussion of the fluvial-aeolian system refers to the aeolian deposits, but also encompasses the source areas, the foreland rivers, the basin, the winds, sufficient aridity/seasonality, orographic blocking, etc – everything that enables the aeolian sediments to be deposited. Our data indicates this was emplaced during the late Miocene. But directly to the reviewer’s point, yes and no. With no appreciable relative uplift in the study area since the late Miocene (aside from a few meters of uplift of the La Pampa Central block relative to the Chadileuvú block and Macachin Basin), the lack of a Pliocene-early Pleistocene record across the study area is quite intriguing. As an underfilled portion of the flexural foreland, one would think if alluvial-fluvial dominated deposition reappeared during the Pliocene, there would be a record of it somewhere. However, with fine-grained aeolian deposits, the preservation potential in the geologic record is quite low, making preservation the exception, not the rule—that was the Meijer and van der Meulen (2023) reference in the original manuscript. Following the idea of preservation potential, either deposition of fine-grained aeolian strata coupled with repeated intervals of erosion, or effectively no accumulation because of aridity, become the most plausible explanations for the lack of Pliocene to early Pleistocene strata in the study area. We lean towards deposition with erosion, rather than zero accumulation because there is certainly no evidence for hyperaridity, but rather drier with more seasonality. There are always exceptions, but generally speaking, the accumulation of fine-grain aeolian deposits requires some vegetation (or significant surface roughness). Of course, we aimed to simplify this in the manuscript so as to not enter into the territory of wild speculation. Given what we have learned from studying wind-eroded bedrock floored basins in China and the probability that the windward margin of the Chinese Loess Plateau has retreated due to wind erosion (e.g., (Kapp et al., 2015; Stevens et al., 2018), and applying it to the idea of the calcrete duricrust in the study area delimiting subsequent (wind) erosion, we tried to present the reader with a credible scenario that is internally consistent with the (current understanding of the) regional geology...which is limited because of the limited Pliocene-early Pleistocene record. Perhaps coupled flux and provenance investigations of dust in the South Atlantic record would shed light on the Pliocene uncertainty.

If the reviewer can think of how we can add clarity for the reader regarding how the Pliocene-early Pleistocene fits into the model, we are open to suggestions.

Lines 266-269: What grain size was ultimately analyzed? Was further sieving performed prior to mounting/analysis? Is there any concern over grain sized biasing when comparing loess-derived zircons to fluvial systems, etc?

AR: We have added to the text the grain size we analyzed – “grains with sizes of silt and very fine sand” Line 310.

The reviewer raises an interesting point in that there can always be some bias based on grain sizes. However, our conclusions are based on the comparison between generally the same-sized grains in upper Miocene strata and Pleistocene-Holocene units. Therefore, therefore concerns are dampened. There is the possibility of some discrepancies between fluvial sediments and the loess we sampled; however, this would not change our primary conclusion in any way (same provenance, same pathways, same fluvial-aeolian system). Additionally, the upper Miocene and Pleistocene-Holocene samples were prepared and analyzed in a self-similar fashion, thus minimizing laboratory-induced biases which could to erroneous interpretations during sample comparisons.

Figure 1: Can you clarify what is meant by loess samples? Those are only Cerro Azul samples, correct?

AR: Correct. To clarify this we added a brief line to the figure caption: Line 336-337.

Figure 2: Would still like to see individual KDEs or CDFs for each sample. Consider labeling y-axis as “# of grains” or similar.

AR: Added the individual KDEs – Figure S1. We altered figure 2, adding Y-axis labels.

Figure 3: At some point it would be great to unpack how these other comparison spectra for SP, Chaco, Patagonia, and Puna were compiled and what geologic context they are from

AR: We agree. However, we are using these data to represent general source areas; there is a risk of overinterpretation at this point in our understanding. We feel the conservative and most prudent use of these data is to use them as general sediment source areas so as to better understand the Pampean aeolian sediments. Further compilations and data sources will allow for additional interpretations.

Figure 4: might label ‘westerlies’ in red text to match arrows.

AR: changed Figure 4 (now Figure 5).

Reviewer #3

General Comments I:

The conclusion that there has been little change in the provenance of sediments within the Pampas since the late Miocene is generally well supported by the new data presented here with a clear and transparent methodology. However, I find that “global climate forcing on the late Miocene establishment of the Pampean system” (as stated in the title) is not currently sufficiently evidenced within the manuscript to support this as the main conclusion of this study. Most of the evidence for this second conclusion seems to come from previous work, and very little of this is shown within this paper (although relevant studies are cited and their conclusions stated in the text). In addition, several of the cited references already make the link between late Miocene aridification and regional sedimentation (e.g. 27-29), so I think a clearer statement of how this study represents a significant advance on what is already known and/or tackles an outstanding question would be beneficial.

Regarding this general comment and comment #III below, we have added a figure (now Figure 4) which shows a synthesis of surface uplift in Asia and South America, the expansion of C₄ ecosystems in Asia and South America, a δ¹⁸O benthic stack, and sea surface temperatures (SSTs). This figure provides a visual reference for the reader to go along with the references which were included in the original manuscript. Additional references were added to the manuscript through this process. In

summary, this figure shows that orographic blocking of westerly sourced moisture to the Pampas was emplaced well before the deposition of the Cerro Azul Fm. Additionally, this figure shows the expansion of C₄ ecosystems in Asia and South America overlaps the depositional age of the upper (i.e., aeolian) Cerro Azul Fm. Finally, and most importantly for our interpretations, the figure shows a pronounced increase in meridional SST gradients between the tropics and mid-latitudes in both hemispheres during the late Miocene when the Cerro Azul Fm. was deposited. This increase in the meridional SST gradients largely resulted from the cooling of the mid-latitudes SSTs. This change occurred in both hemispheres, and because of the robust link between Hadley Circulation and low-to-mid-latitude SST gradients across timescales (e.g., (Bjerknes, 1966; Brierley et al., 2009; Feng et al., 2018; Hong et al., 2023; Li et al., 2023; Lindzen, 1994; Seo et al., 2014; Terray et al., 2003) and many others), this provides quite compelling corroborating evidence for our hypothesis that a shift in the characteristics (i.e., intensity and/or latitudinal extent) of the Hadley cells drove changes in mid-latitude aridity and dust dynamics during the late Miocene.

The focus of this study is on the aridification of the Pampas during the late Miocene. Assuming elevations in the Andes have increased or remained steady over the past 10 million years, which is a safe assumption given all we know of Andean tectonics, there are no viable tectonic processes that could have led to the aridification of central Argentina. Moisture in the Pampas was derived from eastern (South Atlantic Ocean) and northern sources, including the South American Low Level Jet (SALLJ), which transports moisture from the tropics. Whereas the SALLJ is created by the barrier imposed by the Central Andes, increasing surface elevations within the Andes would have only served to increase the SALLJ and thus moisture delivery to central Argentina (Carrapa et al., 2018). Similarly, a rise in elevation in the Argentine Andes would have likely led to increased precipitation (e.g., Insel et al., 2010), which is counter to the observed aridification. Orographic effects would not impact the Pampas because the region does not receive significant precipitation from westerly winds, nor is there any evidence they did so during the late Miocene. There is no question that the Andes play a role in local and regional climate, but there are no viable, documented changes to the Andes during the late Miocene that could explain more arid conditions in the Pampas.

In contrast, changes in Hadley Cell circulation would have been highly effective in causing more arid conditions in the Pampas and surrounding regions during the late Miocene. The so-called late Miocene cooling of SSTs is well documented (Herbert et al., 2016) by a host of paleoclimate indicators. Paleoclimate models of South America indicate this change in late Miocene temperatures would have resulted in contraction, or intensification, of Hadley Cell circulation. The models indicate the intensified Hadley Cells would have reduced moisture delivery to central Argentina (Carrapa et al., 2018). There is plenty of evidence that the region became more arid during the late Miocene, which include the onset of aeolian sedimentation in the Pampas as well as those references cited in the text.

To support the new figure additional text has been added to the Discussion:
Lines 207-228

Regarding the contribution of this work, we argue that the data, interpretations and discussions provided in this manuscript represent a significant advance in our knowledge of South American dust dynamics, wind regimes, and climate more broadly.

This assertion is supported by comments from Reviewers 1 and 2. Firstly, we demonstrate the onset and continuity of the Pampean eolian system, one of the largest continental aeolian systems in the world. Knowledge of when this system began and how it has evolved did not exist prior to these data. Secondly, we show that the onset of this system coincided with regional aridification that can be tied to changes in synoptic circulation during the late Miocene, which again was unknown prior to this study. And thirdly, we tie the onset of the Pampean aeolian system in the southern hemisphere to the development of the Chinese loess plateau in the northern hemisphere. To our knowledge, this is the first attempt to provide a bi-hemispheric hypothesis that brings together the origins of two large continental loess/aeolian provinces. This hypothesis will spring a number of follow-up studies including those in sedimentation, climate modeling, and sediment-climate feedback systems.

General Comments II:

In order to prove that global climate forcing was responsible for the establishment of the Pampean system, I think there are four key points that need to be clearly demonstrated: 1) The Pampas was established in the late Miocene. 2) A dominant local tectonic driver of this event can be ruled out. 3) There was local climatic change at this time that can explain the establishment of the Pampean system. 4) This climatic change can be linked to/explained by global factors.

Regarding the 4 proofs requested by the reviewer:

1) *“The Pampas was established in the late Miocene”*

We altered the text to show that the Late Miocene Cerro Azul Formation contains the first evidence of aeolian strata in the Pampas (Lines 109-110) – hence the establishment of the Pampean aeolian system. We also added that the lower portion of the Cerro Azul Formation is devoid of aeolian deposits.

2) *“A dominant local tectonic driver of this event can be ruled out”*

We added statements to discuss why the tectonic hypothesis is unviable (Lines 207-228). There is no scenario where tectonics in the Andes can produce more arid conditions in the Pampas – orographic barriers of westerly winds were not (and are not) relevant to precipitation in the Pampas. If anything, surface uplift in the Andes would have led to greater precipitation by focusing the SALLJ towards the Pampas (*ceteris paribus*). In contrast, GCM models demonstrate that changes to synoptic-scale climate patterns (e.g., Hadley Cells) are effective in shutting off moisture to the Pampas.

3) *“There was local climatic change at this time that can explain the establishment of the Pampean system”*

Lines 217-220 contains a list of data that indicate local climate change “evidenced in central and southern South America through carbon and oxygen isotope data in pedogenic carbonates and fossil enamel, and fossil crown height” - Carrapa et al., 2018, Hynek et al., 2012, and Bywaters Reyes et al., 2010. In addition, the appearance of eolian strata in the Cerro Azul Formation provides a line of evidence that local climate has changed.

4) *“This climatic change can be linked to/explained by global factors”*

We added that climate models indicate global cooling of sea surface temperatures during the Late Miocene lead to climate change in South America – “Climate models indicate more arid conditions in the late Miocene Pampas and Sierras Pampeanas

resulted from changes to synoptic-scale atmospheric circulation patterns that reduced moisture export from tropical South America”. (Lines 207-215)

General Comments III:

Four points raised by the reviewer:

1) *There are some new ages presented in this paper, which are reported to be consistent with previous work. If these new ages are adding additional value, this could be more clearly emphasised to help strengthen this manuscript. Also, as the timing of loess formation is key to the main conclusions, a figure or panel demonstrating the distribution of reported ages would be helpful. The late Miocene spans ca. 6 million years. Can you be any more precise about the likely timing of Pampas establishment and/or comment on whether it was rapid or gradational?*

2) *The text currently states that the “fluvial-aeolian system is the product of Andean tectonics, which provides a steady influx of sediment and influence lower-level patterns, combined with more arid conditions that emerged during the late Miocene”. Can you comment on how important these two drivers are relative to each other? This is crucial to how strongly the message that global climate was the main driver can be stated but is not currently discussed in detail.*

3) *L190-196 list a range of previously published evidence for regional climate change. This is really important to the key message, and I think the paper would benefit from plotting some of these data (or a summary of their results) in a figure to allow readers to see this evidence for themselves. This could be combined with the age data in point 1.*

4) *Links are made in the text to both late Miocene cooling and the development of the Chinese Loess Plateau in the text, but again, no evidence is directly presented to the reader within this study to demonstrate how similar timings/patterns of change are to justify the strength with which this key message is presented. This could also be shown as part of a figure.*

Regarding the 4 questions/topics raised by the reviewer:

1) *Late Miocene age? Can we be more specific?*

New figure added (now Figure 4 – climate and tectonics synthesis).

At this point there is simply not enough data to be more specific about the age of these units. However, this uncertainty mirrors uncertainties around the expansion of C4 ecosystems in both hemispheres. The most up-to-date data from this region is presented in Previsiti et al., 2021 and is incorporated into our results. Whereas more refined chronostratigraphy would be very welcomed, it is not a part of this study, nor would it alter the main conclusions. Speculating the precise ages of the beds is tempting, but not scientific. We have included our maximum depositional age calculations (MDAs) to provide greater confidence that these beds were deposited during the late Miocene. However, MDAs do not reflect actual depositional ages – they only provide a constraint by giving the oldest age a bed could possibly be. We added some text to clarify this – Line 123-127

2) *Relative role of tectonics?*

Late Miocene moisture in the Pampas was provided by eastern sources (Atlantic Ocean) and northern sources (South American Low-Level Jet) – which is still the dominant pattern today. Tectonics (which we assume means surface uplift in the Andes) would have little impact on the 1st order characteristics of this system. Some uplift could cause more precipitation in the Andes near the Pampas, but overall, Andean tectonics did not play a large role in this area. No moisture is associated with westerly winds in these latitudes so there would not have been any orographic effect.

In summary, tectonics do not appear to have played the primary role. The relative role is still unknown, but is probably best investigated with climate model studies – which are not in the purview of this manuscript.

To address this reviewer's comment, we tried to make this more explicit in the text - (Lines 207-228 and new Figure 4).

3) *Climate data*

We attempted to add these data to Figure 1, but we subsequently removed it because it simply added more material to an already complicated image. Instead, we added a brief sentence to the figure caption (1) stating that the paleoclimate data we reference in the text were collected from the Sierras Pampeanas. – Line 341-342

4) *Link to Loess Plateau*

In an attempt to not relitigate the timing of the establishment of the Chinese Loess Plateau, we provided four key references (Refs. 65–69 [original]) to the reader. We could have selected references from an exhaustive list all showing approximately the same thing (i.e., there was a massive expansion in the loessic deposits on the Ordos Plateau around 7–8 Ma). Why the range one might ask. 1) This is, in part, because the basal loessic sediments were filling in existing topography which had significant relief. To that end, the initial deposition was non-uniform. 2) Like the Cerro Azul Fm., paleontology and magnetostratigraphy are the primary means of depositional age control. Both approaches lack the precision needed to claim the onset of deposition with much precision.

Line-by-line comments

L31: "a global forcing mechanism" is quite vague.

AR: we added "global-climate" forcing mechanism. Line 34

L35-36: This implies that there was no wind-blown dust before the establishment of the Pampas, which seems unrealistic!

AR: Our understanding is that this a teaser, meant to provide a brief overview of the manuscript topic, and not a synopsis of the aeolian history of South America. Furthermore, we do not believe this teaser implies there was never any wind-blown dust prior to this time. However, if this is an issue we can propose an alternative teaser.

L44: Which instances?

AR: We have provided two references that can be read to explore such instances. Line 49

L44-50: There's a bit of a reasoning jump here. Why is S. America in particular "critically important"?

AR: In our view, the logic is clear – South America provides large amounts of wind-blown detritus (and the macronutrients and micronutrients it carries) to the South Atlantic, which could ultimately help sequester CO₂. It may be part of a larger positive-feedback cycle (more dust = colder climate = more glaciers = more dust...). Therefore, it is critical to understand the long-term history of dust and eolian activity in this region.

L58, L64: "western Pampas" v "southwestern Pampas" – is the distinction significant? As a general point, please make figure 1 tie into the text more closely so that a reader unfamiliar with the study area can easily identify the features described.

AR: Good point. We changed this and following instances to make this more consistent. E.g., Line 62

L64-65: Reference?

AR: we have added a reference for this. Line 69

L64-L72: It's not clear what's being stated as already known and what's presented as the novel findings of this study in this paragraph.

AR: As none of this material is cited, it understood to be new information from this study, which will be outlined in the following text – – with the exception of the information related to the Chinese Loess Plateau, which has citations.

L71: "similar mechanism" rather than "thesis".

AR: changed Line 77

L90-92: Phrase this as a hypothesis? Are you trying to set up the idea that large province shifts might still be expected in the latest Miocene and that's something that your data will allow you to explore?

AR: Both Reviewer #2 and #3 had a question about this sentence – so we changed it to make it more evident that it is a statement rather than a subtle question. It is a statement that summarizes our current understanding of how provenance has changed slightly in response to recent tectonic uplift. This has been demonstrated in Garzanti et al., 2022. The primary point of the sentence (beyond the explicit statement) is to make the reader aware that this is a tectonically active foreland and therefore sediment provenance is constantly evolving as crustal blocks are displaced and eroded. Line 96-98

L91: Define bajadas

AR: The term "Bajada" – coalesced alluvial fans – is a common term in geology and physical geography. However, recognizing that this a multidisciplinary journal the text now includes "(i.e., coalesced alluvial fans)". Line 96-97

L93-106: Some repetition here with L58-72.

AR: The former lines (93-106 on the original submission) are a brief introduction to the strata and the area as part of the introduction. These latter lines are part of the "Background" and include a more detailed description.

L103-L104: More arid/greater aridity and/or seasonality than what?

AR: Good point. We altered the text to say '...middle Miocene'. Line 110-111

L108-134: More explanation would be useful throughout the results section to help a non-specialist audience.

AR: In general, we have tried to do this – primarily though the changes requested in the following comments.

L109-110: A comment on the geographical spread of the samples would be useful here.

AR: We added that this is across a 300 km distance Line 118

L110-111: What is/is the purpose of a KDE?

AR: This is more common in the sediment provenance world, but it is basically a smoothed histogram. We include both the histogram and the KDE in the figures. We changed the text to provide some information on the KDE - Line 120-122

L116: What about the lithology of lower half of the Cerro Azul formation? Is this relevant in the aridification history of the region?

AR: Good point. We added some information to the "Background" stating that the lower

half of the formation consists of fluvial sandstones and paleosol horizons (rivers and floodplains) and loess and paleosols in the upper half. Line 103-106. overall, this is consistent with aridification during the late Miocene.

L117: Explain “U-Pb zircon age modes”.

AR: Corrected – we have added “Age populations, or modes...” to help explain this. A mode or population is the term given to groups of detrital zircons with the same ages. Line 128

L138-139: From the same locations? Or averaged across region/exposures?

AR: Added that the data are combined into one KDE. Line 149 We also added “Figure S1” to the list of figures – this figure shows that the individual KDEs are identical to the Pooled KDE. This concern was also raised by Reviewer 2.

L143: “the same” rather than “equivalent”?

AR: Yes. Changed this. There is no reason to avoid the word “same” Line 154

L145: “all samples” stated earlier – do some Pleistocene samples differ?

AR: Changed this to “the samples” – there is also Figure S1 that corroborates this. Line 156

L159-162: The phrasing of this sentence is quite confusing

AR: agreed. Changed this to what we think is easier to comprehend: “MDS plots show the greatest similarity between the Cerro Azul Formation U-Pb data and the Colorado, Negro, and Desaguadero Rivers, which are known to be important nodes in the dust production pathway...” Line 171-175

L163-168: What determines which sources are the most important? Wind/river direction? Erodibility? Extent of exposure? Other?

AR: This is a tricky question and one we as a community are still trying to tease apart. A number of variables play a role – amount of zircons in the source rock (i.e., fertility), erodibility, size of the source area and sediment delivery system, etc. – all of which vary between locations and through time. In short, the large number of uncontrolled variables precludes any reasonable answer to this question, and we are hesitant to speculate in error. However, we do suggest one likely cause for the similarity between the Rios Colorado and Negro and the dissimilarity with the Rio Deseaguadero. There is geological evidence to support the idea that the Colorado and Negro rivers existed in some form during the late Miocene (i.e., they were supplying sediment), whereas most of the evidence suggests the Deseaguadero River did not exist at that time. Therefore it is not surprising that the Cerro Azul Formation has more affinity with the sediment in the Colorado and Negro rivers. Line 175-178

L173-178: This is an example of a section where clearer integration with Fig 1 would be helpful.

AR: This is fair. Figure 1 shows all of the features mentioned in this section, although to someone unfamiliar with the area, it may take some time to absorb the details. We added a reference to Figure 1 in the text Line 190, Line 224. In addition, we added a bit more explanation of the aeolian system to the figure caption of Figure 1 – Line 339-342. We hope that these two changes will make it easier to follow the text.

L183-185: How important are these two factors? References needed for late Miocene aridification.

AR: This is another good question, but one that cannot be quantitatively addressed at this time, and certainly not with this data set. Both are needed – a sediment source and conditions conducive to eolian transport, thus we did not attempt to discuss which may have been more or less important. Added the citation requested – Line 202

L187: Unclear which part of the sentence “possibly” refers to.

AR: We removed ‘possibly’ Line 203-206

L185-189: How do deposition rates change?

AR: Interesting question. We do not address this for two reasons: 1) without adequate chronological constraints, sedimentation rate estimates are of very limited to no use. These units lack the age constraints needed to reconstruct an effective sedimentation rate. 2) This study examines a ~300 x 100 km area. The sedimentation rate is very sensitive to the location selected in the Pampas and central Argentina. In some areas, there are no Pliocene strata preserved, but in other parts (e.g., Colorado Basin) there are relatively thick Pliocene deposits. The location would greatly impact rates.

L204-206: Confusing – “may have” and “it did so” seem to contradict each other. Are you saying it persisted in one location but not extensively?

AR: Corrected – removed “may have” – Line 238-244

L218: Any other lines of evidence to support this change in Hadley cell dynamics?

AR: At this point, not that we are aware of. In this part of the discussion we are trying to present potential new avenues of research.

L237: Three of your 9 samples have MLA >9Ma, so can the formation of the Pampas be directly linked to the 8Ma start of the CLP? This age discrepancy should be acknowledged and discussed.

AR: Maximum depositional ages are relatively imprecise tools that indicate a particular bed has to be younger than X Ma (you cannot have a 10 myr old bed with 8 myr old zircons). They are not necessarily coincident with depositional ages. We added a brief explanation of this to the Results section - “Although these ages represent the maximum possible depositional age, and not necessarily the actual depositional age, they are consistent with existing age constraints” – Line 123-127

L244: Why is the similarity in spatial footprints relevant?

AR: In this section we are attempting to show the similarities between these two continental aeolian systems. An important component of that is to demonstrate they are of the same size.

L249: “would be needed” instead of “had to occur”.

AR: changed to “would have needed” Line 290-291

L255-257: “the global nature of our findings” currently not supported by the presented data

AR: We removed the “global” part of this and altered it to say that the parallels between the Chinese Loess Plateau and the Pampas suggests late Miocene climate changes influenced mid-latitude conditions and deposition. – Line 298-301

L267: Clarify if >500 or <500µm fraction kept.

AR: changed this to show that we kept the <500 µm material. There are virtually no zircons >500 µm. Further processing and separation of such large grains is therefore pointless. For our samples, this was not a problem because most of our grains were <100 µm. This is a standard procedure, but we have changed the text to clarify this. – Line 312

L292-293: Part of sentence missing?

AR: This is not an issue in our draft, but we will certainly fix this if it remains a problem.

L314: “Simplified” not “general” model to avoid any confusion with GCMs?

AR: Excellent point. Text has been changed. Line 377

L324: Please carefully check reference formatting e.g. L350

AR: We have gone back and checked formatting.

L686: Figshare link not currently working, but I assume would be made live upon acceptance?

AR: Correct. An embargo will remain in place on the Figshare folder until this manuscript has finished the peer review process and accepted. All data were provided with this submission.

Figure 1: I find this figure messy and overly complex. I suggest aligning it more closely with the aspects of local geography discussed in the text to help the reader follow the main arguments. Is it possible for the geological units shown to more closely align with the 8 age modes used in the study? What does all of the white space represent? Is the Cerro Azul represented by dots on both white and yellow backgrounds? Figure would look smarter if framed by a box. Also ensure latitude tick marks are outside edge of map (40S appears to be within national boundary).

AR: This is a fair point. We spent a significant amount of time altering Figure 1 so that it is easier to comprehend, but also maintains all of the necessary information. We have simplified the geologic map and muted the colours, which enhances the Quaternary and Miocene aeolian strata as well as the sample locations. We are hesitant to break up the geology to correspond with the age modes because the exposures of age mode rocks may not correspond to all sources – For example, Mesozoic and Cenozoic sedimentary rocks contain a lot of 220-280 Ma zircons, in addition to the exposures of 220-280 rocks. To keep it simple and less biased, we simply show the modern geological exposures.

We added a box around the figure, corrected line widths, and more palatable.

In general, we believe the revised figure 1 is more in line with the reviewer’s request – it lessens the geologic map and enhances the key features that are discussed in the text.

To the figure caption we’ve added that the paleoclimate data discussed in the text is from the Sierras Pampeanas.

Figure 2: Add y-axis labels & ensure tick marks touch axis.

AR: corrected figure 2

Figure 3: Label axes. Need to acknowledge source of background image in panel B?

AR: The axes in MDS plots are typically not labeled because they represent dissimilarity values that are dependent upon the dataset – they are not absolute values, but values

generated to fit the data. We added the citation for the background figure in the figure caption Line 364

Figure 4: Add “aeolian production areas” to legend. Can’t see loessoid deposits anywhere on the map? What is the significance of the different arrow colours? Define SALLJ.

AR: We added these items to Figure 4 - the legend and made the loessoid deposits more prominent. We’ve added the “westerlies” label for the red arrows. Added the SALLJ definition to the figure caption Line 386.

S1: Ensure CaCO₃ formatted correctly throughout. What is “Type 1 Locality”?

AR: These comments refer to the Table S1 in the supplementary files – we have corrected the formatting and we added a statement to note that “Type 1 Locality” refers to the type section of the Cerro Azul Formation as defined in Visconti et al., 2010.

- Bjerknes, J., 1966, A possible response of the atmospheric Hadley circulation to equatorial anomalies of ocean temperature: *Tellus*, v. 18, no. 4, p. 820-829.
- Feng, J., Li, J., Jin, F., Zhao, S., and Zhu, J., 2018, Relationship between the Hadley circulation and different tropical meridional SST structures during boreal summer: *Journal of Climate*, v. 31, no. 16, p. 6575-6590.
- Herbert, T. D., Lawrence, K. T., Tzanova, A., Peterson, L. C., Caballero-Gill, R., and Kelly, C. S., 2016, Late Miocene global cooling and the rise of modern ecosystems: *Nature Geoscience*, v. 9, no. 11, p. 843-847.
- Hong, Q., Feng, J., and Li, J., 2023, Modulation of the Central Pacific El Niño on the Relationship between the Hadley Circulation and Tropical SST: *Journal of Geophysical Research: Atmospheres*, p. e2022JD038195.
- Kapp, P., Pullen, A., Pelletier, J. D., Russell, J., Goodman, P., and Cai, F., 2015, From dust to dust: Quaternary wind erosion of the Mu Us Desert and Loess Plateau, China: *Geology*, v. 43, no. 9, p. 835-838.
- Li, Y., Du, M., Feng, J., Xu, F., and Song, W., 2023, Relationships between the Hadley circulation and tropical sea surface temperature with different meridional structures simulated in CMIP6 models: *Frontiers in Marine Science*, v. 10, p. 1145509.
- Lindzen, R. S., 1994, Climate dynamics and global change: *Annual Review of Fluid Mechanics*, v. 26, no. 1, p. 353-378.
- Seo, K. H., Frierson, D. M., and Son, J. H., 2014, A mechanism for future changes in Hadley circulation strength in CMIP5 climate change simulations: *Geophysical Research Letters*, v. 41, no. 14, p. 5251-5258.
- Stevens, T., Buylaert, J. P., Thiel, C., Újvári, G., Yi, S., Murray, A. S., Frechen, M., and Lu, H., 2018, Ice-volume-forced erosion of the Chinese Loess Plateau global Quaternary stratotype site: *Nature Communications*, v. 9, no. 1, p. 983.
- Terray, P., Delécluse, P., Labattu, S., and Terray, L., 2003, Sea surface temperature associations with the late Indian summer monsoon: *Climate Dynamics*, v. 21, p. 593-618.

REVIEWERS' COMMENTS

Reviewer #2 (Remarks to the Author):

Thanks for making the suggested changes.

Reviewer #3 (Remarks to the Author):

I thank the authors for a thorough consideration of the feedback provided by the reviews to produce a clearer and better argued manuscript. I now recommend this study for publication in Nature Communications.

Minor points

Fig 1 caption: "Chadileuvú block (CB), Pampa Central block (PCB), and San Rafael block (SRB)." Seems to be an incomplete sentence?

Fig 4 & caption: Westerhold et al. 2020 is a benthic isotope composite record, not a stack.

Line-by-line response to reviewer comments

Reviewer #2 (Remarks to the Author):

Thanks for making the suggested changes. The authors thank the reviewer for the measured guidance and handling of the review process.

Reviewer #3 (Remarks to the Author):

I thank the authors for a thorough consideration of the feedback provided by the reviews to produce a clearer and better argued manuscript. I now recommend this study for publication in Nature Communications. Agreed. We thank you for your contribution to improving this manuscript.

Minor points

Fig 1 caption: "Chadileuvú block (CB), Pampa Central block (PCB), and San Rafael block (SRB)." Seems to be an incomplete sentence? Agreed. Lines 341–342 now read:

"Abbreviations are as follows, Chadileuvú block (CB), Pampa Central block (PCB), and San Rafael block (SRB)."

Fig 4 & caption: Westerhold et al. 2020 is a benthic isotope composite record, not a stack. Good catch. Replaced "stack" with "composite record".